# Environmental DNA Metabarcoding in Marine Ecosystems: Global Advances, Methodological Challenges, and Applications in the MENA Region

**DOI:** 10.3390/biology14111467

**Published:** 2025-10-22

**Authors:** Sandy K. Sawh, Sarah Merabet, Nayla Higazy, Marwa Béji, Johan Mølgård Sørensen, Pedro Range, Ahmad M. Alqudah, Mohamed Nejib Daly Yahia

**Affiliations:** 1Environmental Science Program, Department of Biological and Environmental Sciences, College of Arts and Sciences, Qatar University, Doha P.O. Box 2713, Qatar; ss2201550@student.qu.edu.qa (S.K.S.); sm1302497@qu.edu.qa (S.M.); nh1404517@qu.edu.qa (N.H.); bejimarwa@gmail.com (M.B.); cjt429@alumni.ku.dk (J.M.S.); 2Environmental Science Center (ESC), Qatar University, Doha P.O. Box 2713, Qatar; prange@qu.edu.qa; 3Applied Research, Innovation and Economic Development Directorate, University of Doha for Science and Technology, Doha P.O. Box 24449, Qatar; 4Natural History Museum of Denmark, University of Copenhagen, 1353 Copenhagen, Denmark; 5Biological Science Program, Department of Biological and Environmental Sciences, College of Arts and Sciences, Qatar University, Doha P.O. Box 2713, Qatar; aalqudah@qu.edu.qa

**Keywords:** environmental DNA (eDNA), metabarcoding, marine biodiversity, sustainable management, Middle East and North Africa (MENA)

## Abstract

**Simple Summary:**

Marine ecosystems in the Middle East and North Africa (MENA) support high biodiversity but remain understudied due to environmental challenges and limited data. Environmental DNA (eDNA) offers a non-invasive approach to identify species from genetic traces in the water, reducing the need for direct capture. This method has proven effective for detecting rare species, monitoring impacts of pollution, and identifying potential invasive species. Key challenges include incomplete reference databases and the need for region-specific protocols. Combining eDNA with traditional surveys can enhance biodiversity monitoring and support more informed conservation and management strategies.

**Abstract:**

Environmental DNA (eDNA) metabarcoding has transformed marine biodiversity monitoring by allowing non-invasive, cost-effective detection of species with high resolution across diverse marine habitats. A systematic literature search was conducted using Google Scholar, Scopus, and the Qatar University Library databases. Relevant peer-reviewed publications were screened and selected based on predefined inclusion criteria to ensure comprehensive coverage of studies. This review synthesizes advances in global and regional eDNA applications, emphasizing the Middle East and North Africa (MENA) region, which faces unique environmental extremes, high endemism, and significant data gaps. eDNA metabarcoding often outperforms traditional methods under comparable sampling effort to traditional surveys in detecting rare, cryptic, and invasive taxa, but technical challenges like incomplete reference databases, primer biases, PCR inhibitors, and inconsistent methodologies limit their effectiveness, particularly in understudied areas such as MENA. Recent developments, including multi-marker approaches, autonomous samplers, and next-generation sequencing, are enhancing detection precision and enabling broader, real-time monitoring. In the MENA region, early studies have revealed eDNA’s potential for habitat distinction, biogeographic research, pollution assessment, and the early discovery of non-indigenous species, although progress is hindered by gaps in reference libraries, infrastructure, and regulation. This review underscores the urgent need for regional collaboration, standardized protocols, and capacity-building. By integrating eDNA with traditional methods and leveraging emerging technologies, the MENA region can establish itself as a leader in marine biomonitoring under extreme environmental conditions, providing actionable insights for conservation and sustainable management of its unique marine ecosystems.

## 1. Introduction

Environmental DNA (eDNA), defined as genetic material shed by marine organisms into water, sediment, or ice, has transformed biodiversity monitoring by enabling non-invasive, cost-effective species detection across marine ecosystems [1,2,3]. Initially applied to microbial communities in the 1980s, eDNA gained traction in marine science following proven success in freshwater ecosystems [4]. These early studies demonstrated its utility for detecting invasive and endangered aquatic organisms [2,5]. eDNA metabarcoding can detect up to 30% more species than traditional survey methods, including rare and cryptic taxa, but this advantage is conditional on factors such as the completeness and reliability of reference databases, primer performance, and bioinformatic workflow parameters, all of which introduce variability and uncertainty in detection outcomes [6,7,8]. Recent meta-analyses have further confirmed eDNA’s advantage in terms of cost-effectiveness and sensitivity [7,9,10,11].

Advances in high-throughput sequencing have greatly enhanced the utility and scalability of eDNA metabarcoding, enabling rapid, simultaneous analysis of multiple samples for large-scale biodiversity assessments [5,12,13]. Studies have consistently demonstrated the efficacy of eDNA metabarcoding in predicting species richness and detecting habitat segregation across diverse marine ecosystems [2,6,14,15].

While eDNA methods offer transformative potential for marine biodiversity monitoring, several technical and methodological challenges require careful consideration [3,16,17,18,19]. A major limitation is the incompleteness and unreliability of reference databases, which directly impact the accuracy of taxonomic assignments [18,20,21]. For instance, GenBank analyses reveal that only ~50% of Northeast Atlantic marine fish species have reference sequences for the widely used 12S rRNA marker, while mislabeled or erroneous entries further compromise identification [18]. Many regions in the MENA, such as the Red Sea and Arabian Gulf, are biodiversity hotspots that remain under-surveyed, leading to major gaps in species records, creating blind spots for detecting non-indigenous or cryptic species [22].

Primer bias also represents a critical constraint, as amplification efficiency varies across taxa, skewing community composition profiles. Widely adopted primers like Teleo and MiFish (12S rRNA) show inconsistent performance even within the same ecosystem [3,17]. Concurrently, bioinformatic inconsistencies, such as variability in clustering thresholds, denoising algorithms, and taxonomic assignment parameters, further induce variability in species detection rates across studies [19].

Despite these challenges, eDNA workflows offer stronger standardization potential than traditional survey methods. Field sampling and laboratory processing, like water filtration, preservation, and DNA extraction, are increasingly harmonized, reducing/minimizing observer bias inherent in visual censuses. However, cross-study comparability remains restricted by heterogeneous marker selection, sequencing depths, and bioinformatic approaches [23,24]. This does not mean that traditional methods can be discounted; long-term comparisons of underwater visual surveys demonstrate that carefully structured protocols can yield consistent community-level diversity estimates across disparate approaches [25]. Taken together, these results suggest that while eDNA holds greater promise for standardization, pairing it with explicit frameworks and transparent taxon-selection choices is essential to manage trade-offs in detections and comparability.

Ongoing innovations are addressing these limitations. Machine learning algorithms now optimize primer design and correct for amplification biases, improving taxonomic detection accuracy. Meanwhile, initiatives like the *GAPeDNA* project audit the reference database gap. Advances in portable sequencing technologies enable real-time, field-based eDNA analysis, and blockchain-based data sharing platforms ensure reproducibility across studies. These developed tools are repositioning eDNA into a robust and globally applicable tool for marine conservation [26].

Though eDNA methods have rapidly advanced global marine biodiversity monitoring, the application of these techniques within the marine ecosystems of the Middle East and North Africa (MENA) region remains notably underrepresented in the literature. The MENA region encompasses unique and diverse habitats, such as the Red Sea and Arabian Gulf, which are characterized by extreme environmental conditions and high levels of endemism yet suffer from significant gaps in reference databases and baseline biodiversity data. This review particularly examines the progress, challenges, and future directions of eDNA metabarcoding studies in MENA marine environments, highlighting region-specific barriers such as limited taxonomic coverage and harsh sampling conditions while identifying critical research needs to advance biodiversity assessment in these globally essential but understudied ecosystems.

## 2. Current Applications of eDNA Metabarcoding in Marine Ecosystems

Monitoring marine ecosystems requires accurate, scalable, and sensitive methods to assess species composition, biodiversity, and ecological interactions across diverse habitats [27]. Traditional survey approaches, such as visual censuses and net sampling, often face limitations in scalability, detection sensitivity, and invasiveness, especially for elusive or rare taxa. In contrast, environmental DNA (eDNA) metabarcoding has emerged as a transformative, non-invasive tool for marine biomonitoring [6,7,8]. It is increasingly recognized for its efficiency, rapidity, and ability to provide detailed insights into species presence, genetic diversity, and population structure [28,29,30]. Since its introduction to marine biomonitoring [1], eDNA metabarcoding has been widely applied to detect organisms spanning multiple trophic levels, including bacteria, algae, protists, invertebrates, fish, and marine mammals. This approach enables comprehensive biodiversity assessments and facilitates ecosystem-scale monitoring, which are critical for effective conservation and management of marine resources [31,32].

### 2.1. Microbial Communities as Bioindicators

Microbial communities have proven to be significant bioindicators [33] given their rapid response to environmental changes such as pollution, hypoxia, and shifts in temperature or salinity [34,35,36,37,38]. The application of eDNA metabarcoding enables detailed monitoring of microbial diversity and community shifts, providing early warning signals for changes in water quality, nutrient dynamics, and the emergence of pathogens [38,39,40]. Integrating eDNA metabarcoding with established approaches can improve the accuracy and resolution of marine environmental quality assessments, supporting more effective management of reef health. Such studies underscore the utility of eDNA metabarcoding for long-term, cost-effective biomonitoring and environmental/ecological quality assessment [41,42,43].

### 2.2. Coral and Symbiont Detection

Recent advances demonstrate that eDNA metabarcoding is highly effective for detecting both coral species and their symbiotic algal communities in reef environments. Multiple studies have shown that eDNA can reliably identify a broad range of *Scleractinia* corals, including key reef-building genera such as *Acropora*, *Montipora*, *Platygyra*, *Pocillopora*, and *Plesiastrea*, as well as their associated symbionts across diverse locations, using both universal and taxon-specific primers [44,45,46,47,48,49]. The choice between universal and specific primers allows for flexible targeting, balancing broad biodiversity detection with enhanced resolution for the focal taxa. However, eDNA results may include signals from both extracellular and intracellular DNA, which could confound precise assessments of live presence versus detrital or symbiont DNA, highlighting the importance of careful data interpretation in coral reef biomonitoring [44,48,50].

### 2.3. Detection of Elusive, Endangered, and Invasive Species

Monitoring certain marine species, particularly those that are invasive, endangered, or highly mobile, remains challenging with traditional survey methods due to their low abundance, scattered distribution, or cryptic behavior [30]. eDNA metabarcoding offers a rapid, sensitive, and cost-effective alternative for detecting these species, often outperforming conventional approaches in both scope and resolution [3,51].

Numerous case studies highlight the effectiveness of eDNA methods in detecting a broad spectrum of fish species across diverse marine habitats. Applications range from identifying endangered species such as scalloped hammerhead sharks, European eels, and Macquarie perch to tracking invasive redfin perch and documenting rare or locally extirpated species like the sharp-toothed lemon shark and European weather loach [52,53,54,55,56,57,58]. These examples demonstrate the capacity of eDNA methods to reveal both species presence and community composition, even at low organismal densities [49,52,55,59,60,61,62].

Beyond fish, eDNA methods have been successfully applied to a wide range of marine taxa. It has enabled detection of non-indigenous invertebrates, including arthropods, mollusks, and Polychaete worms [63], as well as invasive ascidian species [64] and colonial tunicates [65]. The method has also proven effective for monitoring endangered species such as the sea cucumber *Holothuria scabra* [66] and for assessing native biodiversity [67]. Collectively, these studies underscore the value of eDNA methods as a comprehensive tool for biodiversity monitoring and environmental assessment across trophic levels [64,68,69,70]. As the technique continues to advance, its integration with traditional monitoring approaches will further enhance our ability to track and manage marine biodiversity, including in data-limited regions such as the MENA.

### 2.4. Marine Conservation, Restoration, and Ecosystem Health Monitoring

The ongoing global decline in marine biodiversity is exacerbated by limited knowledge of species distributions and abundances, particularly in complex and understudied ecosystems [71]. Traditional monitoring methods often lack the sensitivity and scalability required to detect rare, cryptic, or invasive species, resulting in significant knowledge gaps. In contrast, eDNA metabarcoding has emerged as a robust and efficient alternative, providing high-resolution data on species presence and community composition across diverse marine habitats [1,60,72]. This approach is rapidly gaining acceptance as a standard tool for marine biomonitoring [73].

eDNA metabarcoding has been successfully applied to a range of marine environments, including coral reefs [44,46,47,48,61,74,75], mangroves [76], seagrass beds [77], and open ocean and deep sea [36,42,62,67,78,79]. Notably, coral reef ecosystems, recognized for their biodiversity and vulnerability, have benefited from eDNA-based approaches, which enable comprehensive and non-invasive assessments of coral and invertebrate diversity, including those inhabiting artificial reef structures used in restoration [44,46,75].

In fisheries science, eDNA metabarcoding is increasingly used for stock assessments and to monitor valuable or rare fish species, often providing more precise abundance estimates than traditional surveys [52,60]. The method also facilitates the detection of shifts in community composition over time and space, offering critical insights into ecosystem resilience and responses to environmental change [58,72]. Such insights are essential for assessing marine ecosystem resilience, environmental recovery capacity, and the impacts of both natural and human-induced activities [29]. Importantly, eDNA metabarcoding supports evidence-based management by informing the design and evaluation of Marine Protected Areas (MPAs), enabling the detection of rare, cryptic, or elusive species that might otherwise go unrecorded [73]. By complementing traditional monitoring and fisheries catch data, eDNA can help guide sustainable resource management and conservation strategies [80].

eDNA metabarcoding has significantly advanced the monitoring and management of marine biodiversity, particularly within reef environments. Its integration with traditional survey methods enhances the detection of a broad range of taxa, including rare, cryptic, and invasive species, while also supporting more accurate assessments of community composition and ecosystem health. Notably, recent applications extend beyond macrofauna to include emerging work on plankton and microbial communities, providing new insights into biodiversity patterns, seasonal dynamics, and ecosystem functioning, particularly for taxa that are difficult to sample using conventional methods. Despite its promise, the application of eDNA in marine settings is challenged by factors such as DNA dilution in large water volumes, variable salinity, environmental dynamics (e.g., currents, tides), and methodological constraints [72]. While this review highlights key advances, it is not exhaustive; the field continues to evolve rapidly, with new applications and methodological improvements emerging. Addressing ongoing technical and ecological challenges will be essential to fully realize the potential of eDNA metabarcoding for comprehensive reef monitoring and effective conservation, both globally and within the MENA region.

## 3. Comparative Analysis of eDNA Methods and Traditional Survey Methods

An increasing number of studies seek to compare traditional biodiversity monitoring methods with eDNA metabarcoding to evaluate their relative effectiveness in marine ecological assessments [78]. Traditional approaches, such as trawl surveys [81], visual censuses [46,52,60,82], and Baited Remote Underwater Video systems (BRUVs) [55], rely on direct species observation or capture the physical identification and classification of species. However, these methods are often limited by phenotypic plasticity, life stage variation, and the cryptic nature of some species [27]. They can also be costly, environmentally disruptive, and less effective in hard-to-access habitats. Trawl surveys, for example, may underestimate diversity due to species avoidance, habitat preferences, and gear limitations [78,83,84,85,86,87].

On the other hand, eDNA metabarcoding offers a non-invasive, rapid, and scalable alternative for biodiversity monitoring with minimal ecosystem disturbance [72]. Comparative studies across various marine habitats have consistently shown that eDNA is reliable for detecting and identifying fish species, as well as estimating their abundance and distribution [72,78]. For instance, a 14-year study on fish community structures in the surface and bottom waters of the Sea of Japan revealed that eDNA metabarcoding identified 112 fish species from 94 eDNA samples, outperforming 140 underwater visual censuses that detected 80 species [60]. eDNA was particularly effective at detecting rare and cryptic species, revealing greater taxonomic diversity than visual surveys [52].

When compared to BRUV, eDNA metabarcoding has also demonstrated higher species detection rates. One study reported that eDNA identified nearly three times as many marine species as BRUV, recommending increased eDNA sampling alongside BRUV to improve community assessments [55]. Similarly, in Indonesia, eDNA detected more shark species than both BRUV and visual surveys, highlighting its efficiency while also emphasizing the value of using multiple methods for robust biodiversity monitoring [82]. In deep-sea environments, eDNA metabarcoding has shown comparable or even superior performance to trawl surveys. Studies in Japanese bottom waters and Greenland found strong agreement between eDNA and trawl data on species richness and composition, with each method detecting unique taxa, underscoring their complementary detection capabilities [60,78,81].

Based on mounting evidence, integrating eDNA metabarcoding with traditional survey methods offers a more comprehensive and accurate approach to marine biodiversity monitoring. At this stage, one method cannot completely replace the other; a combined strategy is recommended to improve detection of both common and elusive species, increase the reliability of biodiversity assessments, and support more effective conservation and management of marine ecosystems. In particular, research in reef environments has shown substantial overlap in species detection between eDNA and visual surveys, highlighting the value of using both methods together to obtain a fuller picture of community composition and ecosystem health [46,60].

## 4. Methodological/Experimental Challenges

The eDNA analysis workflow consists of basic steps, which are sample collection and processing, DNA extraction, and PCR amplification and sequencing, followed by bioinformatics and data analysis and interpretation [88]. (See Appendix A for a breakdown and description of methodological eDNA workflow.) Each step introduces specific technical and ecological challenges that can influence the accuracy, sensitivity, and reproducibility of eDNA-based analysis. In this section, we will examine the challenges associated with each stage of the process.

### 4.1. Sample Collection and Processing

Sample collection is a critical step in eDNA metabarcoding, as it directly influences downstream processing and data interpretation. Unlike other steps in the workflow, sampling is a one-time opportunity, making careful planning and execution essential [89]. Defining the sampling area, selecting unbiased sampling points, and determining the appropriate sample quantity and subsampling strategy are all crucial for ensuring robust and scalable results. These choices affect the statistical power of the study and its ability to detect ecologically meaningful patterns [89].

Standardization of sampling protocols, including area dimensions, sample substrates (such as water, sediment, or biofilms), and collection methods, is necessary to minimize variability between sites. Additional factors, such as pooling subsamples, total sample volume, and water depth, can further influence the sensitivity of eDNA detection and ecological interpretation [17,72,89,90].

Contamination remains a major and significant challenge in eDNA studies, as the introduction of foreign DNA can lead to false positives and misinterpretation of ecological data. One of the most common contamination sources includes field equipment and laboratory reagents. To reduce risk, best practices involve thorough sterilization and decontamination using bleach, ethanol, or UV light, as well as the use of single-use consumables like gloves and filters [89,91].

Filtration is the most widely used method for capturing eDNA from water samples; however, its effectiveness depends on the filter material, pore size, and water quality [88,92]. Cellulose-based membranes often perform well, but smaller pore sizes (e.g., 0.2 µm) can clog in turbid waters, reducing efficiency and potentially introducing PCR inhibitors [93], while larger pore sizes reduce clogging but may lower DNA capture efficiency, requiring site-specific optimization.

Sample storage and transport protocols are critical for maintaining DNA integrity, as degradation can occur within 48 h and is exacerbated by heat, UV exposure, microbial activity, and uncontrolled pH [94,95,96,97]. Experimental evidence shows that frozen samples remain the most stable over long durations, making immediate on-site filtration and freezing the preferred option. Where these are not feasible, alternative strategies include on-site ethanol or isopropanol precipitation, short-term refrigeration or chilling, or the application of chemical preservatives that stabilize DNA until laboratory processing. This tiered approach allows flexibility in handling diverse field conditions while minimizing loss and degradation of eDNA [88,89,92]. See Table 1 for the decision matrix on alternative sample processing.

### 4.2. Molecular Processing and Sequencing Challenges

After sample collection, eDNA studies require several molecular steps, including DNA extraction, PCR amplification, and sequencing. Each stage presents unique technical challenges that can affect detection sensitivity, taxonomic resolution, and reproducibility [31,72,97]. Extracting high-quality DNA from marine samples is often a difficult challenge. Commercial kits, such as the DNeasy Blood & Tissue Kit and the PowerWater DNA Isolation Kit (Qiagen, Hilden, Germany), are widely used but vary in efficiency depending on sample type and context [98]. Environmental factors such as specimen density, water movement, and sediment content can influence DNA yield and quality [99,100]. Additionally, fine sediments may bind or retain DNA, leading to false negatives or, if DNA persists after a species has left, false positives.

Organic matter and various substances in marine environments can inhibit PCR. Humic acids, polysaccharides, phenols, and minerals like calcium can interfere with DNA amplification, resulting in reduced sensitivity or false negatives [100,101]. These inhibitors may become more concentrated during filtration and are particularly concerning in water bodies rich in organic material, such as those containing algae. Polysaccharides, phenols, and acids found in algae can further disrupt amplification by affecting enzymatic reactions or competing with target DNA [102].

The choice of genetic markers and primers is critical for successful eDNA metabarcoding. Common markers to detect genes include mitochondrial genes (COI, 12S, 16S rRNA) for animals, chloroplast loci (rbcL, matK) for plants, and ribosomal genes (16S, 18S, ITS) for microbes [72,103]. While the COI gene is a standard barcode for animals, its long amplicon size and primer design challenges can limit its use in degraded eDNA samples [104]. Shorter markers like 12S and 16S rRNA are often preferred for marine surveys due to higher amplification success and broader coverage [72,105]. For microbes, the 16S rRNA gene is commonly used, but taxonomic bias and incomplete reference databases can limit accuracy [103]. For example, some regions may detect certain groups like SAR11 or Archaea better than others [106], while issues such as variable copy numbers and low taxonomic resolution can lead to underestimation of specific lineages [107]. Primer design also affects detection. Broad-range primers can detect diverse taxa but may lack specificity, while narrow-range primers offer precision but limited coverage [108]. Using a combination of broad and specific primers or a multi-marker approach can help balance these trade-offs. (See Table 2 for a comparative analysis of molecular markers).

High-throughput sequencing (HTS) using state-of-the-art next-generation sequencing (NGS), typically on Illumina platforms (e.g., MiSeq, HiSeq), is the standard for eDNA metabarcoding due to high accuracy and suitability for short amplicons [117]. More recent platforms, like Oxford Nanopore, provide longer reads and improved taxonomic resolution but historically had higher error rates; however, recent advances are reducing these limitations [118]. Multiplexing samples on HTS platforms increases efficiency but introduces risks such as index hopping, amplification bias, and demultiplexing errors. Strategies to address these include dual-indexed primers, two-step PCR protocols, and rigorous quality control [72].

Quality control is essential throughout the workflow. Positive and negative controls help monitor contamination and artifacts. Comparing results to mock communities and using qPCR for uncertain detections further improves reliability. Technical replicates and platform-specific controls are recommended to ensure robust and reproducible results. Standardizing molecular protocols and addressing these challenges are critical for enhancing the accuracy and comparability of eDNA metabarcoding, particularly for effective biodiversity monitoring in reef environments and the MENA region [72,101,119].

### 4.3. Data Analysis and Interpretation Challenges

After sequencing, bioinformatics pipelines are used to process the raw sequence reads into ecological data. This complex workflow involves several critical steps, including quality control and filtering of reads, chimera removal, denoising, and ultimately, taxonomic assignment and downstream analysis [20,72,120,121]. A variety of software tools are available for these tasks, such as QIIME, DADA2, VSEARCH, and OBITools [88]. However, the lack of standardized protocols presents a significant challenge; differences in pipeline design, software selection, and parameter settings can introduce analytical biases, making it difficult to compare results across different studies.

A key decision in eDNA data processing is whether to use Operational Taxonomic Units (OTUs) or Amplicon Sequence Variants (ASVs) for clustering sequences. OTUs group similar sequences to reduce the impact of sequencing errors, but they may not accurately represent true biological taxa and can overestimate diversity [122]. ASVs provide higher resolution by distinguishing unique DNA sequences without arbitrary similarity thresholds, improving reproducibility and comparability across studies. However, cross-study reuse requires strict harmonization by trimming reads to the identical primer-defined amplicon segment, using the same locus and orientation, and fully reporting primer sequences and trimming parameters, or alternatively aggregating results to the lowest common taxonomic rank or using phylogenetic placement when loci differ. ASVs can also capture intra-individual and multicopy variation (e.g., heteroplasmy or multicopy mitochondrial/nuclear targets), so locus choice, curation, and downstream filtering rules need to be carefully calibrated [123,124,125].

Despite these challenges, ASVs are now preferred over OTUs for their greater precision and reliability. For reliable cross-study comparisons and effective batch-effect control, eDNA workflows should use the same primer regions, trim sequences to consistent lengths, and apply unified chimera detection methods. Techniques like LULU filtering and swarm clustering can help minimize the splitting of closely related sequences. Including technical replicates and tracking batch variables improves statistical validity, while standardized rules for data filtering, normalization, and taxonomic assignment help ensure consistency. To support reproducibility and future reuse, raw FASTQ data and sample metadata should be deposited in public archives like SRA, and all analysis scripts and workflow details should be shared on repositories such as Zenodo, OSF, or GitHub [122,126].

### 4.4. Taxonomic Assignment and Database Limitations

The final and most crucial step in the pipeline is the assignment of a taxonomic identity to each sequence, which is dependent on the quality and completeness of reference genome databases. Accurate taxonomic assignment in eDNA metabarcoding relies on comprehensive and high-quality reference databases. After sequence clustering, representative reads are compared to databases such as NCBI GenBank and the Barcode of Life Data Systems (BOLD).

The utility of these public databases is constrained by significant limitations. These databases are not stringently curated; sequence entries may be submitted without mandatory taxonomic validation or supporting voucher specimens. This lack of oversight increases the risk of misidentification and taxonomic errors, particularly for rare or less-studied taxa. Systematic curation and validation of records by trained taxonomists are required to address these shortcomings, yet the declining pool of experts presents a substantial challenge for future regional and global biodiversity research initiatives. Furthermore, many taxonomic groups remain underrepresented or entirely absent from current reference databases, severely limiting the ability to assign eDNA reads to lower taxonomic levels [72,127]. This underrepresentation restricts ecological interpretation, which results in a considerable proportion of eDNA sequences remaining unclassified, which in turn impedes comprehensive biodiversity assessments [18,20,21].

Even when matches are found, errors such as mis-annotations or incomplete metadata in reference entries can result in false positives or ambiguous identifications [128]. These challenges reduce the effectiveness of eDNA for biodiversity assessment and management, particularly in regions with limited baseline data. To address these issues, several countries have launched initiatives to improve reference databases. For example, national projects in Austria, Finland, Norway, the Netherlands, and Germany, as well as the DNAqua-Net consortium, aim to expand and curate aquatic reference sequences for environmental monitoring in Europe [127,129]. In contrast, the MENA region lacks a coordinated effort. Although there are scattered entries in global databases, the region still suffers from a notable absence of species-specific barcodes and molecular references.

Establishing regional initiatives alongside qualified professionals to develop complete DNA databases for the MENA would significantly improve taxonomic resolution and the interpretation of metabarcoding data, ultimately enhancing our understanding of regional biodiversity [45]. (See Table 3 for summary of metabarcoding challenges, solutions, and best practices).

## 5. Successful Applications of Marine eDNA (MENA Region)

Marine eDNA research in the MENA region has demonstrated several advancements within the last five years. Alongside global counterparts, focus has been on developing methods for biodiversity assessment, biogeographic analysis, and environmental monitoring (Figure 1).

Biodiversity Assessment

eDNA metabarcoding has proven effective for characterizing marine biodiversity in the Arabian Gulf. The authors in [58] targeted the mitochondrial 12S rRNA gene to study vertebrate communities across diverse habitats, including seagrass beds, coral reefs, mangroves, and sand bottoms. This approach successfully identified both common and elusive species, such as dugongs and sea snakes, providing a comprehensive picture of biodiversity over a large spatial scale. Similarly, ref. [45] conducted an ecological assessment of the Arabian Gulf using eDNA metabarcoding to analyze benthic community structures. Their findings revealed valuable insights into how these communities respond to environmental stressors, emphasizing the utility of eDNA for monitoring ecosystem health.

Biogeographic Pattern Analysis

Researchers have employed eDNA metabarcoding to investigate biogeographic patterns along the Omani coastline by targeting multiple taxonomic markers, such as 18S rRNA for eukaryotes and ITS regions for corals and sponges [130]. Their study identified a well-documented biogeographic break in fish communities between northern and southern Oman, correlating these shifts with local environmental factors and anthropogenic impacts. Comparably, ref. [114] explored benthic bacterial and eukaryotic communities along a crude oil spill gradient in a Persian Gulf coral reef. This research demonstrated how pollution gradients influence community composition, further validating the effectiveness of eDNA for detecting environmental changes.

Monitoring Non-Indigenous Species

The introduction and spread of non-indigenous species (NIS) have been effectively monitored using eDNA techniques in the MENA region. Ref. [131] conducted the first assessment of biofouling assemblages in the Red Sea using eDNA metabarcoding. Their study highlighted how maritime traffic contributes to NIS proliferation, emphasizing the importance of eDNA for early detection and monitoring in data-limited regions.

Marine Habitat Differentiation

eDNA has also been utilized to differentiate habitat types based on sediment quality and ecological conditions. Ref. [132] assessed benthic foraminifera as proxies for environmental quality in Kuwait Bay, demonstrating how eDNA can reveal distinct habitat signatures associated with sediment pollution and degradation. In addition, [133] evaluated toxic sediment effects around Abu Ali Island, Saudi Arabia, using eDNA profiling to assess how pollutants influence biotic communities. (See Table 4 for a MENA Region Study inventory.)

### 5.1. Methodological Innovations in Marine eDNA Research (MENA Region)

Recent advancements in eDNA methodologies have significantly enhanced the accuracy, efficiency, and scope of biodiversity assessments in marine environments. These innovations encompass a range of approaches, from multi-marker methods to autonomous sampling devices and novel sequencing techniques, each addressing critical challenges in marine research.

The use of multiple gene regions for biodiversity assessment has emerged as a powerful methodological innovation. The authors of [58] demonstrated the effectiveness of targeting both mitochondrial 12S rRNA and COI genes to capture a broader range of marine taxa, including vertebrates and invertebrates, across various habitats in the Arabian Gulf. This multi-marker approach provided a more comprehensive characterization of marine communities compared to single-marker methods, underscoring its utility for large-scale biodiversity studies. On the other hand, ref. [45] employed eDNA metabarcoding to analyze benthic communities in the Arabian Gulf, highlighting how primer design and marker selection influence detection accuracy and ecological interpretations.

Autonomous sampling devices, such as the “Fish Sensing Box,” represent a significant innovation by enabling offshore sampling over extended periods without requiring constant human presence [58]. These devices are particularly valuable for studying temporal dynamics in marine biodiversity and detecting rare or transient species. By reducing the logistical challenges associated with traditional sampling methods, autonomous devices facilitate large-scale monitoring programs. However, their deployment may be constrained by cost and technical complexity, especially in resource-limited regions. Improved habitat-specific sampling methods have further enhanced eDNA detection across diverse environments. For example, sediment sampling has proven effective for assessing benthic communities as proxies for environmental quality, as demonstrated by [132] in Kuwait Bay. Similarly, ref. [114] utilized sediment samples to analyze bacterial and eukaryotic communities along a pollution gradient, showcasing how tailored approaches improve detection sensitivity and ecological interpretations.

Direct DNA sequencing has emerged as a potential alternative to traditional metabarcoding as it eliminates the PCR amplification step. This method sequences all DNA within a sample and uses advanced bioinformatic tools to extract taxonomic information, avoiding biases associated with primer selection and amplification efficiency [131]. Direct sequencing holds potential for providing more comprehensive biodiversity data, particularly in complex ecosystems such as coral reefs and oil-polluted areas [114,133]. However, its implementation requires significant computational resources and expertise in bioinformatics, posing challenges for laboratories in regions like the Middle East and North Africa (MENA).

### 5.2. Ecological Insights from Marine eDNA Research (MENA Region)

One study in the Arabian Gulf has demonstrated the ability of eDNA metabarcoding to differentiate marine vertebrate communities across diverse habitats such as seagrass beds, coral reefs, mangroves, and sand bottoms [58]. Their research revealed fine-scale spatial differences and habitat-specific eDNA signatures, providing a more nuanced understanding of biodiversity distribution, including rare species like dugongs and sea snakes. Similarly, ref. [132] used sediment eDNA to assess benthic foraminifera in Kuwait Bay, uncovering distinct ecological patterns influenced by sediment quality and pollution levels. These findings highlight the utility of eDNA for detecting localized environmental impacts and habitat-specific community structures. Additionally, ref. [114] examined pollution gradients in Persian Gulf coral reefs using eDNA profiling, identifying significant shifts in bacterial and eukaryotic community composition linked to oil contamination. These studies collectively demonstrate how eDNA methods can uncover ecosystem responses to anthropogenic impacts, providing valuable insights for conservation planning and environmental management in data-limited regions.

### 5.3. Challenges and Limitations in Marine eDNA Research (MENA Region)

Marine eDNA studies in the Middle East and North Africa (MENA) region face several challenges and limitations that are closely linked to the area’s unique environmental conditions, infrastructural constraints, and regulatory frameworks. Addressing these challenges is essential to fully harness the potential of eDNA systems for biodiversity monitoring and conservation in the region.

Environmental Challenges

The arid climate and transient nature of many water bodies in the MENA region result in high turbidity levels, particularly during dry seasons. These conditions necessitate tailored sampling techniques, such as employing multiple filters for smaller volumes or implementing staged pore-size filtering to reduce initial turbidity [58]. Additionally, high temperatures and intense solar radiation accelerate the degradation of eDNA, potentially affecting detection rates and sample quality. For example, studies like those conducted by [130] emphasize the importance of optimizing sampling timing and preservation methods to maintain sample integrity under such challenging conditions. These environmental factors underscore the need for adaptive methodologies tailored to the region’s climatic extremes.

Technical and Infrastructural Limitations

A major challenge is the lack of comprehensive genetic/genome reference libraries for local species, which restricts the ecological interpretation of eDNA data [22,134]. Without well-curated databases, researchers face difficulties in identifying species accurately and developing cost-effective biomonitoring programs. Furthermore, there is often a shortage of trained personnel and specialized laboratories equipped for eDNA analysis in some MENA countries. This gap in infrastructure leads to logistical hurdles in sample processing and data interpretation, slowing the progress of regional eDNA applications.

Regulatory and Standardization Issues

Marine eDNA research in the MENA region is further complicated by regulatory fragmentation across countries. Each nation develops its own frameworks and standards for environmental monitoring, leading to inconsistencies in methodologies and reporting requirements. This lack of harmonized regulations makes it challenging to compare results across studies or implement region-wide monitoring programs. Moreover, environmental monitoring regulations are still evolving in many MENA countries, with intricate reporting requirements that can be difficult for researchers to navigate [77]. A further limitation is the lack of accredited reference laboratories experienced in marine eDNA, which impedes methodological standardization and quality assurance.

This is exacerbated by increasing restrictions on the international shipping of DNA samples, which present logistical barriers and hinder timely analyses, especially in countries requiring import permits and special handling procedures for genetic materials. These combined challenges highlight the urgent need for collaborative efforts toward harmonized regulatory frameworks and standardized protocols. Establishing unified operational standards and reference laboratory networks is essential to facilitate reliable, reproducible, and efficient marine eDNA research in the region.

### 5.4. Global Impact of Marine eDNA Research

The global application of eDNA techniques for monitoring marine biodiversity has expanded rapidly, but significant disparities exist in its implementation across different regions. While eDNA studies are well-established in temperate and arctic areas, particularly in Europe and North America, tropical and subtropical regions, including the MENA, remain underrepresented. These differences underscore both the potential and the challenges of adopting eDNA methodologies on a global scale.

#### 5.4.1. Regional Disparities in eDNA Applications

Globally, terrestrial eDNA research has been concentrated in temperate regions, with 42% of studies conducted in Europe and 12% in North America. In contrast, only 32% of peer-reviewed eDNA biodiversity monitoring studies have been carried out in tropical and subtropical areas, including the MENA region [135]. A similar trend of regional concentration is observed in marine eDNA research, where studies are also predominantly conducted in temperate and well-resourced regions, with comparatively fewer studies in tropical and underrepresented marine areas [135]. This disparity reflects the earlier establishment of eDNA methodologies in temperate systems and the unique challenges faced by tropical and subtropical environments. For instance, high turbidity levels, accelerated DNA degradation due to high temperatures, and a lack of comprehensive reference libraries for local species hinder the widespread adoption of eDNA techniques in regions such as the MENA [58,130].

Despite these challenges, tropical regions hold immense potential for eDNA applications due to their high biodiversity and the limitations of traditional monitoring methods in such ecosystems. For example, UNESCO’s pioneering eDNA program has successfully mapped nearly 4500 marine species across 21 World Heritage sites using standardized sampling methods. This initiative highlights how eDNA can provide rapid and cost-effective biodiversity assessments even in data-limited regions [136].

#### 5.4.2. Advancements in Global eDNA Monitoring

Globally, innovative applications of eDNA have revolutionized biodiversity monitoring. In Latin America and the Caribbean, for instance, eDNA has been used to establish shark diversity under varying levels of anthropogenic impact, monitor tropical reef fish populations, and validate invasive species detection [137]. Similarly, studies in Namibia have demonstrated the utility of tailored sampling techniques in overcoming the high turbidity levels typical of arid environments [138]. These examples emphasize how region-specific adaptations can enhance the applicability of eDNA techniques.

Moreover, technological advancements have facilitated global-scale biodiversity assessments. For example, UNESCO’s Ocean Biodiversity Information System (OBIS) systematically uploads eDNA data to ensure open access for researchers worldwide. This platform has facilitated the comparison of biodiversity data across regions and promoted international collaboration in marine conservation efforts [136].

### 5.5. Summary of Findings

Marine eDNA research has emerged as a transformative tool for biodiversity monitoring and ecosystem management, offering unparalleled insights into ecological patterns across spatial and temporal scales. In the MENA region, studies have demonstrated the utility of eDNA in revealing habitat-specific biodiversity signatures [58,132], identifying biogeographic boundaries shaped by environmental gradients and human activities [130], and tracking ecosystem changes over time [77]. These applications demonstrate eDNA’s capacity to detect fine-scale ecological dynamics, such as shifts in community composition resulting from pollution [114] or the introduction of non-indigenous species [131]. Despite its potential, challenges persist in interpreting eDNA data under complex environmental conditions, where factors like water movement, salinity, and temperature can affect DNA persistence and detection accuracy [132,133].

High turbidity levels accelerated DNA degradation due to extreme temperatures, and salinity gradients complicate sample integrity and detection accuracy [133,139]. Additionally, the lack of comprehensive genetic reference libraries for local species hinders ecological interpretation and limits regional applications. Infrastructure gaps, such as limited molecular biology laboratories, further exacerbate logistical hurdles. Addressing these issues requires investment in capacity-building initiatives, international collaboration, and the development of harmonized methodologies tailored to local conditions. Nevertheless, emerging innovations like autonomous sampling devices [58] and direct DNA sequencing without PCR amplification [131] offer promising solutions for overcoming these barriers, paving the way for more effective biodiversity monitoring and conservation efforts globally.

However, systemic barriers in the MENA region limit the widespread adoption of eDNA technologies. The lack of comprehensive genetic reference libraries remains a critical bottleneck for accurate species identification, necessitating coordinated efforts to expand databases specific to regional biodiversity [45]. Additionally, infrastructural gaps such as limited access to trained personnel and specialized laboratories underscore the importance of capacity-building initiatives to support eDNA research. Addressing these challenges through international collaboration, infrastructure investment, and standardized protocols will be essential for unlocking the full potential of eDNA in the MENA region. Despite these limitations, marine eDNA offers a robust framework for advancing conservation efforts in the data-limited areas, providing critical tools for understanding and mitigating anthropogenic impacts on marine ecosystems.

## 6. Management Applications and Future Research

The MENA region’s extreme marine environments, characterized by hypersalinity and seasonal temperature fluctuations, demand innovative approaches to biodiversity monitoring and policy reform. Environmental DNA (eDNA) metabarcoding has demonstrated higher accuracy in detecting Gulf species assemblages, outperforming visual surveys for cryptic species [45,58]. To move from innovation to impact, three interconnected pathways focused on data and standards, capacity-building, and policy application are recommended to guide regional research and management.

An essential first step is to establish consistent data and methodological standards for eDNA research. Developing and distributing a comprehensive MENA eDNA SOP by 2026, covering sampling protocols, preservation strategies, primer selection, bioinformatics workflows, and quality controls, will help harmonize approaches across projects and countries. Establishing at least three regional biodiversity baselines by 2027 will provide the foundational datasets needed for both research and regulatory decision-making. Building a regional consortium, modeled on UNESCO’s Marine World Heritage framework, to standardize barcode markers (such as 12S rRNA and COI) and address primer biases for extremophile species further supports robust data generation and reliability.

Just as standards are vital, regional capacity-building is required to bridge infrastructural and technical gaps. An expanding network of certified reference laboratories, with at least two regional hubs and a minimum of ten partner labs passing proficiency tests, will foster inter-institutional collaboration and ensure technical competency. The introduction of regular inter-laboratory mock trials will reinforce data quality, create shared learning environments, and enable more rapid troubleshooting of unforeseen challenges.

Policy application relies on translating eDNA results into actionable management. Piloting eDNA integration in real-world scenarios such as marine protected area health assessments, early warning detection of non-indigenous species in ports, and industrial baseline or rehabilitation monitoring for oil, gas, and salt works will establish clear decision pipelines from data collection to archival, sharing, and management action. These applications showcase the value of eDNA in supporting efficient, evidence-based responses to pressing ecological threats.

### Future Marine Research Perspectives for the MENA Region

The extreme physicochemical conditions of the Gulf demand methodological refinement for marine eDNA studies. Elevated temperatures and salinity accelerate eDNA degradation, especially during summer months, and warrant more frequent sampling intervals and tailored preservation protocols [132]. UNESCO’s eDNA Expeditions spanning 21 marine World Heritage sites provide a global framework for comparative studies and offer valuable reference data for future investigations of coral thermal adaptation, including corals in the Gulf’s uniquely harsh environments.

Despite recent advances, current eDNA-based research in the Gulf predominantly targets megafauna and commercially relevant fish, often overlooking microbial and invertebrate communities that underlie fundamental biogeochemical cycles. Addressing this ecological gap is essential for holistic characterization of marine ecosystems and for capturing ecosystem resilience and function in the face of environmental stressors.

Emerging technologies are poised to transform marine biomonitoring in the MENA region. Next-generation sequencing and high-quality genome sequencing are enhancing taxonomic resolution and functional inference from eDNA data. The development of region-specific marine genomic reference databases will be crucial for accurate species identification and ecological interpretation, overcoming current limitations arising from low representation of Gulf taxa in global repositories.

In parallel, the integration of advanced artificial intelligence, robotics, and miniaturized environmental sensors enables high-throughput, real-time acquisition of phenotypic and genetic data directly from marine environments. AI-driven approaches can efficiently analyze large, complex datasets, offering unprecedented capacity for the detection, classification, and prediction of biodiversity trends and ecosystem changes.

Establishing a dedicated Gulf eDNA Observatory to centralize long-term operations, technology deployment, and data integration would strengthen collaboration between researchers, policymakers, and industry stakeholders. Such a multidisciplinary initiative is essential to bridge research-policy gaps and deliver actionable insights for the sustainable management of the Gulf’s unique marine ecosystems. We propose a three-phased strategy: in the short term, standardize sampling protocols, build regional capacity, and launch pilot eDNA monitoring of key species; in the medium term, develop a genomic reference database, expand monitoring to microbes and invertebrates, and incorporate AI-driven analyses; and in the long term, establish a permanent observatory with continuous high-throughput monitoring, using eDNA data to inform policy, spatial planning, and conservation across the MENA region.

## 7. Conclusions

Environmental DNA metabarcoding has emerged as a transformative tool for marine biodiversity monitoring, offering unique sensitivity, scalability, and non-invasiveness compared to traditional approaches. This review highlights global advances and the growing body of research applying eDNA across diverse marine ecosystems, while emphasizing the unique challenges and opportunities within the Middle East and North Africa region. Despite significant methodological progress, persistent gaps in taxonomic reference libraries, standardization, and regional capacity continue to limit the full potential of eDNA applications in these environments. Developing regional databases, harmonized protocols, and cross-institutional collaborations will be critical to improve data comparability and reliability. By integrating eDNA with conventional monitoring techniques and leveraging emerging technologies, MENA countries can enhance biodiversity surveillance, inform conservation strategies, and strengthen sustainable marine resource management under rapidly changing environmental conditions.

## Figures and Tables

**Figure 1 biology-14-01467-f001:**
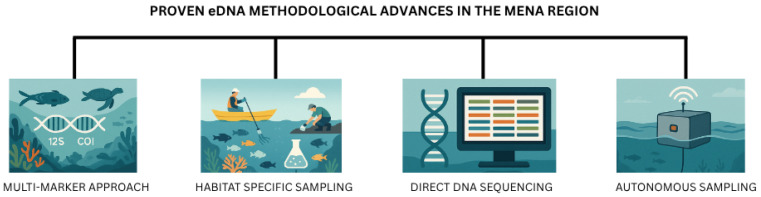
Successful applications of eDNA metabarcoding in the MENA region.

**Table 1 biology-14-01467-t001:** Decision matrix for sample collection and preservation.

Strategy	Description	When to Use
On-site Filtration and Freezing	Filter samples immediately, and then freeze; preserves DNA longest	Ideal when portable freezers available
Ethanol/Isopropanol Precipitation	Add alcohol to samples on-site for chemical stabilization	No freezing available, work in remote sites
Short-term Refrigeration/Chilling	Store samples chilled (4 °C) for up to 24–48 h to slow degradation	Short transport, timely lab access
Chemical Preservatives (e.g., Longmire’s and CTAB)	Use stabilization buffers or solutions for temporary preservation	Fieldwork constraints, delayed processing

**Table 2 biology-14-01467-t002:** Comparative review of molecular markers used in marine eDNA metabarcoding.

Marker Gene	Taxonomic Coverage	Resolution	Database Availability	Advantages	Limitations	References
*12S rRNA*	Vertebrates (esp. fish, mammals)	Species, sometimes genus	GenBank, MitoFish	High specificity for vertebrates, short fragment (good for degraded DNA), popular primers (MiFish, Teleo)	Lower coverage for invertebrates, regional gaps, incomplete reference for non-fish taxa	[5,14,18,60,109,110,111,112]
*16S rRNA*	Broadly targets marine fish and some invertebrates. Provides moderate taxonomic resolution for metazoans and high resolution for prokaryotes	Genus, sometimes species	GenBank, SILVA, RDP	Broad detection of prokaryotes is helpful for microbial community profiling—ability to amplify a range of metazoans, especially invertebrates	Poor animal/fungal resolution, primer bias. Does not perform as well as 12S for fish diversity, incomplete reference databases which limit species-level identification	[5,6,14,20,32,113]
*18S rRNA*	Universal eukaryotes (protists, metazoa, algae)	Family to species, mainly genus	GenBank, SILVA, PR2	Comprehensive eukaryotic coverage, widely used for plankton, protists	Short variable regions limit species resolution, primer bias, not ideal for metabarcoding animals	[14,42,45,82,114]
*COI*	Metazoa (invertebrates, fish)	Species but limited	GenBank, BOLD	High resolution for animals, standard barcode for metazoans	Variable success (e.g., low amplification in some taxa), incomplete databases for marine species	[18,20,21,82,115,116]

**Table 3 biology-14-01467-t003:** Summary of workflow challenges, proposed solutions, and best practices.

Workflow	Common Issues	Proposed Solutions/Best Practices	References
Sample Collection and Processing	- Contamination (field/lab equipment, reagents) - Variability from site selection, substrate, and sample volume - Loss/degradation during storage and transport (temp, UV, pH, time)	- Standardize area, depth, substrate, volume, and replicate design - Randomize or stratify site selection - Rigorously sterilize equipment and use single-use consumables - Implement field blanks - Filter samples immediately or preserve by rapid freezing, chilling, or chemical preservatives - Minimize storage duration before extraction	[72,89,91,92,94]
Filtration	- Filter clogging (e.g., turbid waters) - Inefficient DNA capture - PCR inhibitors (e.g., humic substances) - DNA loss with large pores - Risk of cross-contamination	- Optimize filter material and pore size per environment - Pre-filter or use a mesh for high turbidity - Minimize filtration time - Use sequential filtration or membrane changes for large volumes - Handle filters carefully to prevent contamination - Employ inhibitor removal steps as needed	[88,92,93]
DNA Extraction	- Low yield or degraded DNA (especially with sediment/organic-rich matter) - Variation among extraction kits - Co-extracted inhibitors causing false negatives or positives	- Use validated/commercial kits optimized for sample type (e.g., for water/sediment) - Maximize lysis (e.g., filter shredding, bead beating) - Incorporate inhibitor removal (e.g., clean-up columns, commercial inhibitor kits) - Minimize loss by using efficient elution techniques	[98,99,100]
PCR Bias	- Primer bias - Inconsistent amplification across taxa; environmental inhibitors (humic acids, polysaccharides) - Non-target and off-target amplification - Variable success across marker genes	- Apply multi-marker and multi-primer approaches for broader taxon coverage - Use high-fidelity, inhibitor-resistant polymerases - Add inhibitor removal steps; optimize PCR conditions (touchdown PCR, DNA dilution) - Perform technical replicates	[3,17,19,104]
Bioinformatics	- Variability in software, clustering thresholds (OTU vs. ASV), denoising, and taxonomic assignment - Lack of standardized pipelines hinders cross-study comparability	- Use standardized, transparent, and reproducible analysis pipelines - Adopt ASV-based methods for higher resolution - Benchmark with mock communities and reference datasets - Report software, parameters, and workflow details	[19,23,120,122]
Reference Databases	- Incomplete or mislabeled entries - Low species/taxon coverage, especially for understudied regions - Missing or ambiguous metadata - Bias towards well-studied areas/groups	- Continuous curation/cleaning - Validation of taxonomic identifications - Regional and global initiatives to generate and expand barcode libraries - Routine gap analysis - Standardized metadata and taxonomy - Coordinated efforts for understudied regions	[18,21,45,128]

**Table 4 biology-14-01467-t004:** MENA Region Study inventory.

Country	Habitat	Marker(s)	n (Samples)	Gear/Devices	Key Findings	Data Deposited
Qatar	Seagrass, coral reefs, mangroves, sand bottoms	12S rRNA	21 sites sampled across 2 years	Water filtration	Mapped vertebrate biodiversity, found elusive species (dugongs)	Dryad Digital Repository GenBank
Oman	Coastal, reefs, lagoon, rocky coast, mangroves	16S rRNA ITS2	120 samples from 15 sites	Water filtration, sediment grab	Revealed biogeographic break, multi-taxa/environmental patterns	Dryad Digital Repository OBITools Platform
Saudi Arabia (NEOM, Red Sea)	Lagoon, ports, biofouling substrates	COI, 18S rRNA	39 panels, 70 water samples	PVC panels, water filtration	20 non-indigenous species (NIS) were recorded and detected 71% of all NIS/cryptogenic species	BOLD Systems European Nucleotide Archive (ENA)
Kuwait	Benthic, estuarine bay	18S rRNA	46 surface sediment samples	Modified grab (surface sediment)	73% agreement in environmental quality assessment between morphological and molecular data	ENA Global Biodiversity Information Facility (GBIF)

## Data Availability

No new data were created or analyzed in this study. Data sharing is not applicable to this article.

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
