# Peer review of "Environmental DNA Metabarcoding in Marine Ecosystems: Global Advances, Methodological Challenges, and Applications in the MENA Region"

_biology, 2025, doi:10.3390/biology14111467_

Round 1

Reviewer 1 Report

Comments and Suggestions for Authors
  1. [L1–L4] Revise the title to: Environmental DNA Metabarcoding in Marine Ecosystems: Global Advances, Methodological Challenges, and Applications in the MENA Region.

  2. Simple Summary should be streamlined into 4–5 sentences, avoiding redundancy.

  3. [L38] The phrase “consistently outperforms” should be revised to “often outperforms under comparable sampling effort” to avoid overgeneralization.

  4. The claim of “30% more species detected” and the subsequent note on method dependency should be merged into one statement, clearly presenting conditions and sources of uncertainty (e.g., incomplete databases, primer bias, workflow parameters).

  5. In Table 1, reference formatting should be consistent with the rest of the manuscript; please verify carefully.

  6. [L418] should be labeled as Table 2 because there is already a Table 1 earlier. Also, [L343] (See Error! Reference source ???) is a technical error—please carefully check and correct such mistakes.

  7. [L288–L309] There is repetition of “Immediate on-site filtration and rapid freezing are recommended…” appearing at least twice with similar wording. These should be merged into one concise paragraph, while providing an alternative options matrix (on-site filtration/precipitation/short-term refrigeration/chemical preservation).

  8. The MENA section’s narrative of “research gaps—extreme environments—data deficiency” is clear, but it should conclude with a list of actionable items (e.g., high temperature/salinity/turbidity → sampling and preservation windows; database gaps → local barcoding initiatives).

  9. The discussion of “environmental dynamics (currents/salinity) + methodological constraints” should be moved to the end of the section as “limitations and countermeasures”.

  10. [L362–L383] OTU vs ASV: Add considerations for cross-study reuse and batch effect handling (e.g., alignment by identical primer regions, post-processing with LULU/Swarm, unified chimera thresholds). Also recommend depositing raw FASTQ files and scripts (Zenodo/OSF/GitHub) to improve reproducibility.

  11. In the Coral and Symbionts section, highlight general vs. specific primer combinations as a transferable methodological takeaway, and caution that results may mix extracellular and intracellular DNA signals.

  12. The strategic initiatives should be presented as three pathways with milestones:

  • Data and Standards: Release MENA eDNA SOP v1.0 (covering sampling, preservation, primers, bioinformatics, controls) before 2026; establish ≥3 regional baselines by 2027.

  • Capacity and Network: Build ≥2 regional reference labs, certify ≥10 partner labs through proficiency tests, and conduct two inter-laboratory mock trials annually.

  • Policy and Application: Pilot in three scenarios—MPA health assessments, NIS early-warning in ports, and industrial baseline/rehabilitation monitoring (oil/gas/salt works)—with clear data-to-decision pipelines (archival → sharing → management action).

  1. Use a unified numbered referencing style (MDPI/Biology default). Do not mix author–year citations in the text. Ensure that all quantitative claims (detection rates, costs, “30% more species” etc.) are supported with precise sources (systematic reviews or meta-analyses preferred). Add citations to key methodological and regional studies (e.g., standardization, ASV pipelines, long-read sequencing, reference library initiatives, port NIS surveillance) to strengthen arguments. Verify the spelling of proper nouns (e.g., Sigsgaard, Sørensen, Teleo, MiFish, DNAqua-Net, OBIS). Standardize terminology: environmental DNA → eDNA; shotgun/direct DNA sequencing → metagenomics (PCR-free); and spell out acronyms like BRUVs at first use.

  2. Two additional figures/tables are recommended:

  • Figure: Methodological decision tree/timeline — Sampling → Preservation → Marker choice (12S/16S/18S/COI decision nodes) → Amplification/sequencing → ASV pipeline → Database → QC checkpoints; with a right-hand timeline noting “1980s microbes → 2010s marine applications → 2020s autonomous sampling and nanopore sequencing.”

  • Table: MENA eDNA study inventory — Country | Habitat | Marker(s) | n(samples) | Gear/Device | Key findings | Data deposited? (OBIS/ENA/GBIF). This table should align with Section 5 cases (e.g., refs [59], [47], [123], [126], [125]).

Reviewer 2 Report

Comments and Suggestions for Authors

Dear Authors

This manuscript addresses a highly relevant and timely topic, given the growing global interest in monitoring marine ecosystems using eDNA-based approaches. The subject matter is significant, and I believe this work has strong potential to contribute to the field and be widely cited in future research. Overall, the manuscript is well-prepared, logically structured, and supported by sound discussion.

That said, I believe several minor revisions could further enhance the clarity, completeness, and overall impact of the paper. My specific comments and suggestions are outlined below:

  • Abstract: Please include a concise but clear description of the methodology to allow readers to quickly understand how the study was conducted.
  • Keywords: Avoid repeating words or phrases already included in the title. Consider selecting keywords that broaden discoverability, such as methodological approaches, study region, or target taxa.
  • Introduction: Strengthen the justification for focusing on the Middle East and North Africa (MENA) region. Provide scientific or ecological reasons, such as unique biodiversity, emerging conservation challenges, or limited previous research in this region.
  • Future Marine Research: Based on the findings, I recommend that the authors outline actionable recommendations for stakeholders—such as government agencies, policy makers, and local communities—on how to adopt eDNA-based monitoring. It would be beneficial to organize these into short-term, medium-term, and long-term strategies to provide a roadmap for future research and implementation.

I hope these suggestions will be helpful to the authors in refining the manuscript.

Best regards,

Reviewer 3 Report

Comments and Suggestions for Authors

The manuscript "Environmental DNA Metabarcoding in Marine Ecosystems: Global Advances, Methodological Challenges, and Regional Applications in the MENA Region"  is an example of a good review that is interesting and informative to read. The authors have collected and analysed a large number of references on the issue. The manuscript provides a detailed, clear and well structured description of the problems of using eDNA metabarcoding in marine ecosystems. The manuscript highlights the lag in research on the Middle East and North Africa region, as well as the limited amount of material in genetic databases. In my opinion, this article will be extremely useful for beginners in the field of genetics.

I would like the authors to make just a few minor corrections. I will present them below.

L. 266 with each stage of the process summarized in Table 1. instead of "with each stage of the process (Summarized in Table 1)."

L. 294 unfinished sentence.

L. 343 the reference is missing here.

L. 495 extra round bracket.

Please see the attachment for more detailed comments.

Round 2

Reviewer 1 Report

Comments and Suggestions for Authors

The manuscript has been substantially improved after revision. The quality of writing, data presentation, and discussion have all advanced considerably, and I now find the study suitable for publication

Author Response

Academic Editor comments

Please, clarify line 345 (page 9):

"...balance  these trade-offs. (See 错误!未找到引用源。 for a comparative analysis of molecular markers)."

Change oriental characters by a clear reference.

Authors’ reply to Academic Editor

Clarification of Line 345 (Page 9) formatting error has been corrected, and a clear reference has been included.